# Metformin Prevents Endothelial Dysfunction in Endometriosis through Downregulation of ET-1 and Upregulation of eNOS

**DOI:** 10.3390/biomedicines10112782

**Published:** 2022-11-01

**Authors:** Ana Filipa Martins, Ana Catarina Neto, Adriana Raquel Rodrigues, Sandra Marisa Oliveira, Cláudia Sousa-Mendes, Adelino Leite-Moreira, Alexandra Maria Gouveia, Henrique Almeida, Delminda Neves

**Affiliations:** 1Department of Biomedicine-Experimental Biology Unit, Faculty of Medicine of the University of Porto, 4200-319 Porto, Portugal; 2Instituto de Investigação e Inovação em Saúde (i3S), 4200-135 Porto, Portugal; 3Cardiovascular R&D Centre-UnIC@RISE, Department of Surgery and Physiology, Faculty of Medicine of the University of Porto, 4200-319 Porto, Portugal

**Keywords:** cardiac fibrosis, endometriosis, endothelial dysfunction, inflammation, metformin

## Abstract

This study aimed to evaluate if the treatment with metformin affects the morphologic structure, endothelial function, angiogenesis, inflammation and oxidation-responsive pathways in the heart of mice with surgically induced endometriosis. B6CBA/F1 mice (n = 37) were divided into four groups; Sham (S), Metformin (M), Endometriosis (E) and Metformin/Endometriosis (ME). The cross-sectional area of cardiomyocytes was assessed after Hematoxylin–Eosin staining and fibrosis after Picrosirius-Red staining. ET-1, nitric oxide synthases-iNOS and eNOS, and VEGF and VEGFR-2 were detected by immunofluorescence. Semi-quantification of ET-1, eNOS, VEGF, NF-kB, Ikβα and KEAP-1 was performed by Western blotting. MIR199a, MIR16-1, MIR18a, MIR20a, MIR155, MIR200a, MIR342, MIR24-1 and MIR320a were quantified by Real-Time qPCR. The interaction of endometriosis and metformin effects was assessed by a two-way ANOVA test. Compared with the other groups, M-treated mice presented a higher cross-sectional area of cardiomyocytes. Heart fibrosis increased with endometriosis. Treatment of endometriosis with metformin in the ME group downregulates ET-1 and upregulates eNOS expression comparatively with the E group. However, metformin failed to mitigate NF-kB expression significantly incremented by endometriosis. The expression of MIR199a, MIR16-1 and MIR18a decreased with endometriosis, whereas MIR20a showed an equivalent trend, altogether reducing cardioprotection. In summary, metformin diminished endometriosis-associated endothelial dysfunction but did not mitigate the increase in NF-kB expression and cardiac fibrosis in mice with endometriosis.

## 1. Introduction

Endometriosis, characterized by the presence of endometrial tissue outside of the uterus, is a gynecological disease estimated to affect up to 10% of women of reproductive age [1,2].

In recent years, evidence that links endometriosis to additional health issues emerged [1], which supports the notion that endometriosis is a chronic systemic disease [3]. Moreover, epidemiological data indicate that endometriosis patients present an increased risk for cardiovascular disease (CVD) [4], including major cardiovascular events [5]. This is partly explained by the hormonal treatments or hysterectomy/oophorectomy [5], but endometriosis-associated conditions such as chronic systemic inflammation [6] and imbalance between oxidative species and anti-oxidative defenses (oxidative stress) [7] also independently contribute to CVD onset and progression [8]. Independently of the hormonal treatments, endometriosis patients present a hypercoagulable status that, whilst supporting the healing of the endometriotic lesions, contributes to the occurrence of thrombo-embolic events and tissue fibrosis [9]. Regarding what concerns the influence of hormonal therapies on the CVD risk in patients with endometriosis, it was recently demonstrated that dienogest therapy or oral contraception for less than 3 years does not increase the atherosclerosis risk. On the other hand, a positive correlation was found between the duration of oral contraception and the risk for CVD in women with endometriosis [10]. No evidence of deleterious effects on heart after gonadotropin-releasing hormone analogues or antagonists treatment was reported [3].

Molecular pathways dysregulated in CVD are also affected in endometriosis; vascular dysfunction is associated with upregulation of endothelin-1 (ET-1) production [11] that intervenes in endometriosis-altered pain response [12]. On the other hand, the protective role of endothelial nitric oxide synthase (eNOS) is impaired in endometriosis by increased expression of the eNOS inhibitor asymmetric dimethylarginine (ADMA) [13]. NF-kB activation leads not only to the onset of CVD [14], including atherosclerosis [15], but also to endometriosis-associated inflammation [16]. Angiogenesis, fundamental for the growth of endometriotic tissue [17], also plays a role in atherosclerosis plaque formation [18], with vascular endothelial growth factor (VEGF) appointed, in both processes, as an essential contributor.

MicroRNAs (MIRs) are small non-coding RNAs that post-transcriptionally regulate gene expression by targeting mRNAs [19]. MIRs dysregulation plays an important role in the control of the molecular pathways that underlie endometriosis-associated CVD. Levels of MIR18a, MIR199, MIR342 and MIR310a that regulate cell survival and cardiac regeneration processes increase in the blood of patients with endometriosis [20,21,22,23,24,25,26]. On the other hand, MIR16-1, MIR155 and MIR24-1 [27] have been reported to target elements of the angiogenic pathways. Other MIRs have been identified in the regulation of inflammatory and oxidative stress responses, such as MIR20a [28] and MIR200a [29], respectively.

Current treatments for endometriosis consider hormone-based medication, analgesics and, ultimately, surgical excision of endometriotic lesions. Unfortunately, those interventions do not alter the course of the disease and the discontinuation of conceiving comes with the recurrence of symptoms [30]. In addition, a hysterectomy is far from being infallible, with up to 15% of patients remaining with persisting and 3–5% with worsened pain [31].

The failure of the currently used treatments encourages the identification of alternative pharmacological strategies for the control of endometriosis. Metformin, a biguanide used in the treatment of diabetes, including gestational diabetes, with recognized effects in the control of CVD, emerged as a potential medication for the treatment of endometriosis. Hepatic glucose production by 5′ adenosine monophosphate-activated protein kinase (AMPK)-dependent and independent mechanism is the main target of metformin. AMPK activation promotes autophagy and inhibits the mammalian target of rapamycin (mTOR), a major regulator of cell growth and proliferation [32,33]. Metformin also presents pleiotropic effects, which supports a potential benefit for the treatment of endometriosis [34]. Metformin, through molecular pathways not yet fully established, presents anti-inflammatory, antioxidant, anti-steroidogenic and anti-angiogenic effects that together mitigate the progression of the disease [35,36,37,38,39]. In fact, metformin has been shown to cause the regression of lesions in a rat model of endometriosis [36] and to mitigate inflammation and expression of angiogenesis-related genes in human ectopic endometrial cells [40]. Studies in women with endometriosis treated with metformin are scarce, but it was demonstrated that treatment incremented endometrial receptivity of minimal/mild endometriosis in a study that only enrolled five patients [41]. Moreover, a reduction in inflammatory cytokines in the blood, dysmenorrhea, pelvic pain, dyspareunia and an increment in the percentage of pregnancy after 6 months of metformin therapy was demonstrated [42].

In this study, we aimed to evaluate the role of metformin in the mitigation of CVD risk associated with endometriosis. Pathways related to endothelial function, angiogenesis, inflammation and oxidative stress were characterized in the heart of a mice model of endometriosis. Morphologic changes and expression of the main miRNAs involved in endometriosis and cardiac damage were also identified.

## 2. Materials and Methods

### 2.1. Animal Model

Thirty-seven female B6CBA/F1 mice, first-generation from C57BL/6J males and CBA/J females mating, chosen for their tolerance to tissue-transplantation, were used in the experiments and kept at i3S (Instituto de Investigação e Inovação em Saúde, Porto, Portugal) animal facility. All procedures were performed by blinded trained staff and approved by the Ethical Committee of i3S and the Portuguese competent authority DGAV (Direção-Geral de Alimentação e Veterinária) for animal experimentation, with the internal reference 2018/27. All animal procedures followed the 2010/63/EU directive of the European Parliament and adhered to the ARRIVE 2.0 guidelines. During experiments, mice were housed 4/cage with freely available water and standard rodent chow in 12 h light/dark cycles in a temperature and humidity-controlled room (21 °C and 50%, respectively).

At the age of eight weeks, mice were randomly divided into four groups according to the intervention to be performed: S—sham-operated mice (n = 6), M—metformin-treated sham-operated mice (n = 8), E—endometriosis-induced mice (n = 14) and ME—metformin-treated endometriosis-induced mice (n = 9). Sample size calculation was based on the resource equation method [43] that indicates an n of 4–6 animals per group as appropriate. In order to guarantee an n = 6 in groups E and ME, endometriosis was induced in a higher number of mice. Since the process was successful in all of them, a higher n was obtained in those groups. The surgically induced endometriosis model was achieved by implanting a~2 mm^3^ uterus fragment in either side of an incision in the peritoneum, collected from a longitudinally opened uterus from a donor of the same mouse strain. The endometrium was sided to the peritoneum, according to the method described by Cohen et al. [44] adapted by Neto et al. (data not published). Animals were anesthetized with volatile isoflurane (5% for induction and 2–3% for maintenance), monitored by cardiac and respiratory frequency and treated with buprenorphine (1 mg/kg subcutaneous twice daily) on the surgery day and in the 72 h after. The implants were visualized by ultrasound (Micro Ultrasound Vevo 2100) 4 weeks after surgery. Mice from groups S and M were subjected to a sham operation consisting of a suture on both sides of the peritoneum. Four weeks after the intervention, animals in groups M and ME started the treatment with metformin (Alpha Aesar Chemicals, Haverhill, MA, USA) diluted in the drinking water at a dose of 50 mg/kg/day that was maintained for 3 months. Water consumption was monitored during experiments to ensure no differences in dosage between cages. Animals were euthanized by cardiac puncture under deep isoflurane anesthesia, and the hearts were collected and divided into two equal parts. One half was frozen in liquid nitrogen and stored at −80 °C for molecular analysis, and the remaining half was immediately fixed in 10%-buffered formaldehyde and embedded in paraffin for histological analysis and MIRs quantification.

### 2.2. Computer-Assisted Histologic Study of the Heart

Serial sections (3 μm thick), deparaffinized through incubation in a stove at 60 °C (30 min) and then in xylol (10 min), were hydrated in a series of incubations (5 min) in a decreasing series of ethanol aqueous solutions (100%, 90% and 70% v/v) and water. For each animal, 4 sections were stained with Hematoxylin–Eosin (HE) (Hematoxylin H and Eosin Y 1% v/v alcoholic, Biognost, Zagreb, Croatia) to assess cardiomyocyte area, and another 4 with 0.1% w/v Picrosirius Red (Direct Red 80, Sigma Aldrich, St Louis, MO, USA), to assess myocardial fibrosis.

Sections were visualized by an experienced and blinded individual in an optical microscope (Leitz Wetzlar—Dialux 20, Wetzlar, Germany) equipped with a camera (Olympus XC30, Tokyo, Japan). A cross-sectional area of 50 orbicular cardiomyocytes was measured per animal using Cell B software (Olympus), and eight fields were analyzed with Image-Pro Plus 6 software (Media Cybernetics, Rockville, MD, USA); the ratio of fibrotic area to total tissue area, corresponding to the percentage of interstitial fibrosis, was calculated.

### 2.3. Dual Immunolabelling in Heart Sections

Deparaffinized and hydrated 5 μm-thick heart sections were incubated in 1% w/v Sudan Black B in 70% v/v ethanol aqueous solution (20 min) to reduce autofluorescence, followed by two washes with phosphate-saline buffer (PBS), two with 0.02% v/v Tween-20 in PBS (PBS.T) and water.

The sections were then incubated in 1 M HCl aqueous solution (30 min) for antigen retrieval, neutralized in 2% w/v sodium borate (5 min), washed with water and PBS, blocked with 1% w/v bovine serum albumin (BSA) and 10% v/v normal swine serum diluted in PBS (1 h) and incubated with a mixture of primary antibodies diluted in blocking solution overnight in a wet chamber at 4 °C (Table 1). Negative controls were prepared with the omission of primary antibodies.

The next day, sections were incubated with the appropriate fluorescence-labeled secondary antibodies diluted in 0.1% v/v Triton X-100 PBS (PBS.Tx) (Table 1) in a wet and dark chamber (1 h) at room temperature after washing with PBS/PBS.T. Sections were then washed with PBS.Tx and PBS and nuclei stained with 4′,6-diamidino-2-phenyl-indole (DAPI) (5 min) before mounting in 50% v/v glycerol/PBS and visualization in an Apotome fluorescence microscope (Carl Zeiss AxioImager Z1—Carl Zeiss, Oberkochen, Germany) equipped with a digital camera (AxioCam MRm—Carl Zeiss).

### 2.4. Western Blotting

Heart fragments were homogenized in lysis buffer pH 7.6 (50 mM Tris-HCl, 10 mM NaCl, 5 mM EDTA, 2 mM β-glycerophosphate, 0.25% v/v Triton X-100) supplemented with proteases inhibitors (1:200 v/v) and phosphatases inhibitors (1:250 v/v) (Sigma-Aldrich). Protein concentration was determined using the Bradford Reagent colorimetric assay (Bio-Rad Protein Assay Dye Reagent Concentrate, Bio-Rad Laboratories, Hercules, CA, USA); 25 μg of total protein/sample diluted in 4x gel loading buffer (50 mM Tris-HCl pH 8.8, 10% v/v glycerol, 2% w/v sodium dodecyl sulfate (SDS), 2 mM EDTA and 0.0125% w/v bromophenol blue) with 0.1 M v/v dithiothreitol were loaded in the gels. Proteins were separated by SDS-polyacrylamide gel electrophoresis (SDS-PAGE) using a discontinuous buffer system (Bio-Rad Laboratories) under a constant current of 15 mA/gel until stacking was completed, followed by 25 mA/gel. The concentration of acrylamide in the resolving gel was either 10%, 12% or 14% (Table 2), according to the molecular weight of the studied proteins. The proteins were then transferred to a nitrocellulose membrane with a pore size of 0.45 μm (Bio-Rad Laboratories) in a transfer system (Bio-Rad Laboratories) at a voltage of 30 V (90 min). Blots were stained with Ponceau S, and the images were acquired in a ChemiDoc XRS+ system (Bio-Rad Laboratories) to use as the loading control for the proteins analyzed in the corresponding lanes. Then, membranes were washed in a 0.1% v/v Tween-20 in tris-buffered saline solution (TBS.T) and blocked for 1 h in 5% w/v dried skimmed milk (Molico^®^). After washing in TBS.T, membranes were incubated for 2 overnights at 4 °C with the primary antibodies (Table 2) diluted in TBS.T with 5% w/v BSA and 0.1% v/v NaN_3_. Membranes were washed in TBS.T and incubated with the appropriate horseradish peroxidase (HRP)-linked secondary antibody diluted in 5% w/v dried skimmed milk in TBS.T (1 h) followed by additional washes in TBS.T. Finally, membranes were reacted with Clarity Western ECL substrate (Bio-Rad Laboratories), and protein bands images were acquired in the ChemiDoc.

The intensity of bands was quantified by densitometry using the Image Lab^®^ software (Bio-Rad Laboratories), and semi-quantification of each protein band was calculated through the ratio of the intensity of the band relative to the total stain of the proteins in the respective lane stained with Ponceau S. In order to correct differences due to variations in the exposure time between membranes, the average value of the sham group in each membrane was used as an internal control. Each experiment was repeated four times and averaged per sample; at least 6 different samples/group were used in each analysis.

### 2.5. Real-Time qPCR

Five 15 μm sections were cut with a fresh microtome blade from each formalin-fixed paraffin-embedded block (n = 6–7/group) and placed in RNase-free microtubes. Paraffin was dissolved with xylene at 50 °C (Fisher Scientific, Merelbeke, Belgium), ethanol (Millipore, Darmstadt, Germany) and airdrying of the remaining pellet before incubation with a protease at 50 °C (2 h) and 70 °C (15 min) to remove protein contamination.

RNAs were extracted using the RecoverAll™ kit (Thermo Fisher, Waltham, MA, USA) according to the manufacturer’s instructions. Purified RNA was converted to cDNA using MystiCq™ MicroRNA cDNA synthesis mix (Sigma-Aldrich). Two negative controls per experiment were performed with the omission of Poly(A)Polymerase and of reverse transcriptase.

MIRs were amplified in 96 well plates after mixing 1 µL of cDNA with 1.2 µL of miRNA-specific primer (miScript Primer Assay, Qiagen, Hilden, Germany) (Table 3), 0.25 µL of Universal Primer (Sigma-Aldrich) and 6 µL of SYBR Green (ThermoFisher Scientific). Real-time polymerase chain reaction (RT-PCR) was conducted on the StepOnePlus Real-Time PCR System (Applied Byosystems, Bedford, MA, USA) and started with denaturation (10 min) at 95 °C, followed by 45 cycles of 15 s denaturation at 95 °C, 30 s annealing at 55 °C and 30 s extension at 60 °C. Each PCR was normalized with a control, RNU6-1, using the formula 2^−(CT target–CT control)^ to quantify MIR expression levels.

### 2.6. Statistical Analysis

Statistical analysis was made using GraphPad Prism version 8.0.2 (GraphPad Software Inc. San Diego, CA, USA).

A regular two-way ANOVA was applied to evaluate the effect of the combined factors, endometriosis and metformin. In addition, *post-hoc* analyses using Tukey’s test were performed. Following ANOVA, to elucidate the role of metformin treatment, multiple *t*-tests by row were conducted by assuming consistent scatter and correction by the Holm–Sidak method.

Results are presented as mean ± SEM, and a *p*-value < 0.05 was considered statistically significant. All measurements were relativized to the expression of the sham group by dividing their value by the mean of the sham group.

## 3. Results

### 3.1. Endometriosis Increases Cardiac Fibrosis and Metformin Increases Cardiomyocyte Cross-Sectional Area

Cardiomyocyte cross-sectional area was increased in group M comparatively with the other experimental groups (Figure 1a–d,i); such increment was of 24% in comparison with group S (*p* < 0.001) and of 14% relative to both groups E (*p* < 0.01) and ME (*p* = 0.02). Groups E and ME showed a trend towards increased cardiomyocyte area when compared with group S (*p* = 0.13 and *p* = 0.19, respectively). The two-way ANOVA revealed a significant interaction between the effects of metformin and endometriosis on cardiomyocytes size (F (1.33) = 15.00, *p* < 0.001).

Moreover, regarding cardiac fibrosis computer-assisted analysis (Figure 1e–h,j), the two-way ANOVA showed that endometriosis had a significant effect on the ratio of the fibrotic area (F (1.31) = 3.00, *p* = 0.02), considering that both groups with endometriosis, E and ME, revealed a 56% (*p* = 0.03) and 57% (*p* = 0.04) higher ratio of the fibrotic area than the S group. Interestingly, mice treated with metformin showed a tendency to increase fibrosis comparatively with controls (*p* = 0.12).

### 3.2. ET-1 and VEGF Are Expressed in the Vascular Smooth Muscle Cells and VEGFR-2, eNOS and iNOS in the Endothelium

Figure 2 depicts the dual immunolabeling of iNOS and ET-1, eNOS and α-smooth muscle actin (α-SMA), a marker of smooth muscle cells (SMC) and VEGF and VEGFR-2, in heart sections of mice from the different experimental groups and respective negative controls (Figure 2m,n). ET-1 (red) expression was detected in the cytoplasm of vascular SMC in all groups (Figure 2a–d). Low co-localization of ET-1 and iNOS (green) was observed in vascular SMC (yellow) (Figure 2a–d); iNOS was almost exclusively detected in the endothelium as a thin labeled layer surrounding the vascular spaces. An apparent increase in iNOS labeling was observed in the heart of mice treated with metformin independently of endometriosis (Figure 2b,d). The endothelial isoform of NOS, eNOS (green), was identified in endothelial cells and endocardium of all experimental groups, with an apparent higher labeling than the observed for iNOS (Figure 2e–h). No co-localization with vascular SMC (red) was verified.

VEGF (red) was detected in the cytoplasm of vascular SMC and myocardium adjacent to the endocardium in every group (Figure 2i–l), with an apparent higher expression in group M (Figure 2j). On the other hand, the VEGF main receptor, VEGFR-2, was identified in the endothelium, endocardium and some vascular SMC without apparent differences among experimental groups.

### 3.3. Endometriosis Increases NF-kB Expression in the Heart and Metformin Attenuates Endothelial Dysfunction through ET-1 Downregulation and eNOS Upregulation

The expression of ET-1, eNOS, VEGF, NF-kB, Ikβα and KEAP-1 was evaluated by Western blotting. For each protein, a band with the expected molecular weight was identified: ET-1—24 kDa, eNOS—140 kDa, VEGF—20–25 kDa, NF-kB—65 kDa, Ikβα—39 kDa and KEAP-1—60–64 kDa. Figure 3 shows representative blots for each protein (Figure 3a) and the graphical representation of protein expression relative to the sham group (Figure 3b–g). Respective membranes stained with Ponceau S are displayed below the blots (Figure 3a).

A two-way ANOVA analysis evidenced a statistically significant effect of the interaction of endometriosis and metformin treatment on ET-1 expression [F (1.28) = 7.94, *p* < 0.01]; when performing multiple *t*-tests per row, a decrease in ET-1 expression in group ME comparatively with group E (*p* = 0.03) was found. Group E showed a trend of increased expression relative to group sham (*p* = 0.07). No differences between groups S vs. M (*p* = 0.50) nor M vs. ME (*p* = 0.50) were observed, reinforcing the absence of effect on ET-1 expression when metformin treatment or endometriosis were considered alone.

Likewise, a statistically significant interaction between endometriosis and metformin treatment was found regarding eNOS expression (F (1.28) = 8.73, *p* < 0.01); an increase was found in group ME, relative to groups E (*p* < 0.01) and M (*p* = 0.03). No differences in groups E and M relative to S were observed (*p* = 0.70 and *p* = 0.50, respectively).

VEGF expression presented a decreasing trend in group ME compared to E (*p* = 0.14).

An increase in NF-kB expression was verified in both experimental groups with endometriosis, E and ME [F (1.25) = 7.23 *p* = 0.01]; no differences were observed between M and S groups (*p* > 0.99), neither between ME and E groups (*p* > 0.99), supporting the absence of metformin effect. The expression of the NF-kB inhibitor, Ikβα, had an increasing tendency in group ME relative to group E (*p* = 0.16), but the two-way ANOVA analysis failed to show an effect of both factors, either isolated or associated.

KEAP-1 expression did not vary between groups.

### 3.4. Endometriosis Downregulates Expression of Cardioprotective MIR199a, MIR16-1 and MIR18a

The expression of MIR199a, MIR16-1, MIR18a, MIR20a, MIR155, MIR342, MIR200a, MIR24-1 and MIR320, assessed by RT-PCR and relativized to the expression of group S (1 arbitrary unit), is depicted in Figure 4.

Among the studied MIRs, expression of MIR199a, MIR16-1 and MIR18a were decreased in the heart of mice with endometriosis: MIR199a (F (1,28) =4.60, *p* = 0.04), MIR16-1 (F (1.24) = 4.70, *p* = 0.04), MIR18a (F (1.29) = 5.80, *p* = 0.02); MIR20a almost reached a significant decrease (F (1.28) = 4.20, *p* = 0.05) (Figure 4a–d).

The two-way ANOVA analysis did not demonstrate variation in MIR155, MIR342, MIR200a and MIR24 expression levels, neither in endometriosis nor after metformin treatment (Figure 4e–h). However, expression levels of those MIRs presented a decreasing tendency in groups E and ME, comparatively with groups of mice without endometriosis. MIR320a expression showed a tendency to decrease in both treated groups, M and ME (Figure 4i).

## 4. Discussion

Until recently, endometriosis had been considered a strictly pelvic disease, but now it is evident that endometriosis is a systemic condition that affects multiple organs [3] and that endometriosis-associated systemic inflammation increases CVD risk [16].

Overt CVD is often reported in women with endometriosis later in their lives [45]. In younger women, there is no evidence of structural vessel alterations, but endothelial dysfunction (ED), which antedates the development of atherosclerosis progressing to CVD, has been recognized as an early manifestation of endometriosis [45].

Reduced expression of eNOS and enhanced activity of ET-1 are important changes in ED; ET-1 is a potent vasoconstrictor and platelet activation inducer, whereas decrement in the expression of eNOS results in NO production inhibition. Together, they contribute to endothelial damage and subsequent CVD [11].

In the current heart study, when endometriosis animals were compared to sham counterparts, an antiparallel variation in the expression of ET-1 and eNOS was observed, suggesting that such mechanisms of endothelial damage are operating in endometriosis [13]. Interestingly, the circumstance that heart changes result from a process originating in a distant (pelvic) location provides support for an extended, systemic nature of endometriosis.

In addition, our data show the upregulation of inflammatory pathways in the experimental condition. When comparing groups E and S, there is an increase in NF-kB expression in the heart of mice with surgically induced endometriosis. NF-kB is an important mediator of metabolic and age-induced myocardial disease and a contributor to myocardial inflammation [14] and to the onset of CVD. Its involvement in endometriosis-related inflammation is thus likely, and since these are long-term conditions, the inflammatory process favors the establishment of local structural changes.

In line with NF-kB expression variation, endometriosis led to increased cardiomyocyte cross-sectional area and heart fibrosis. While the changes may be part of tissue repairing in the beginning, in which cell hypertrophy could be the outcome of a compensatory process, later, tissue fibrosis indicates fibroblast activation, impairment of cardiac function and further contribution to the pathophysiology of CVD [46].

These findings suggesting harmful effects on endothelial cells and on the cardiomyocytes may be the underlying mechanism of endometriosis-associated CVD risk, loss of cardiac function and actual CVD observed in women with endometriosis.

In this setting, one reasonable hypothesis is that the anti-inflammatory known effects of metformin might be useful by playing a protective role on CVD risk in endometriosis. In the current study, in endometriosis animals, metformin administration reversed ET-1 and eNOS expression in the heart in a statistically significant fashion. This is a most interesting finding because it suggests that upon metformin administration, early vascular changes are likely to be prevented, and the CVD-prone condition of endometriosis may be stalled.

However, metformin failed to reduce NF-kB levels in the whole heart extracts of mice with endometriosis since no differences were observed between groups E and ME. Furthermore, no differences between mice treated with metformin (M) and controls (S) were seen. It was not an expected result, considering the reported anti-inflammatory effect of metformin, recently reviewed by Bai and Chen [47], including the downregulation of NF-kB expression in the liver of mice [37] and TNF-α-dependent NF-kB inflammatory signaling in cultured hepatocytes [37]. The absence of the effect of metformin on NF-kB expression does not preclude its modulatory effect in the heart. In fact, in cultured breast cancer cells, metformin did not change total NF-kB levels, although a decrement in its nuclear localization and DNA binding ability was noticed [48].

However, such changes in cell localization or the likelihood of sex- or tissue-specific modulation may underlie the unexpected observation that metformin administration to the healthy animals (M) resulted in a significant increase in the size of cardiomyocytes and a slight increase in fibrosis, possibly related to the hypertrophic effect of the treatment [49]. In fact, cardiac hypertrophy can be a physiological or pathological response that depends on the stimulus and on the signaling mechanisms activated by the cardiac stressor [50].

Sustained activation of Akt/mTOR pathways leads to pathological hypertrophy, but AKT1 seems to also be crucial for a physiological response [50]. As an inhibitor of the Akt/mTOR pathway, we expected metformin to reduce cardiac hypertrophy, as previously reported in both mice and humans [51,52]. In line, novel evidence indicates that metformin mitigates cardiac hypertrophy through epigenetic modification and energy regulation [53,54], but that was not the case. We should consider, however, the fact that metformin reduces estrogen secretion in the ovary [39] and that estrogen confers cardioprotection and delays cardiomyocyte hypertrophy [49]. Most of the studies are conducted in male rodents, but the assessment of cardiac metabolism and longevity in female mice demonstrated a reduction in longevity and cardiac fibrosis in metformin-treated animals compared with controls [55]. In order to elucidate the role of metformin in hypertrophic response, including sex-specificity and dose-dependent patterns, additional studies of cardiac contractility and molecular pathways involved in hypertrophic response ought to be performed.

Similarly to the data related to NF-kB expression, we found that although endometriosis increased heart fibrosis, metformin treatment failed again to mitigate those changes. Our hypothesis is that the endometriosis-associated structural changes in cardiac muscle are irreversible or, at most, take a longer time to revert.

The growth of endometriotic tissue depends on angiogenesis [17], and the formation of new vessels from pre-existent vasculature. VEGF, whose levels increase not only in endometriotic tissue but also in peritoneal fluid of women with endometriosis, is an essential contributor to angiogenesis [56]. The role of angiogenesis in CVD is unclear, considering that the neo-formation of vessels is important in post-ischemic lesion recovery but also plays a role in the development of atherosclerosis plaques [18]. We did not see differences in heart VEGF expression among experimental groups. The absence of differences was an unexpected finding, considering that endometriosis is a pro-angiogenic condition and that metformin was appointed as a medication that mitigates the pro-angiogenic environment [36]. However, VEGF expression has never been studied in the cardiac tissue of mice with endometriosis. Thus, the importance of VEGF in the progression and treatment of CVD in mice with endometriosis remains to be clarified [18].

The overproduction of reactive oxygen species (ROS) is also part of the pathogenesis of endometriosis and of ischemic heart disease. The tissue responses to the increase in oxidative stress are complex and involve several molecular pathways. Among those pathways, activation of Nrf2, a transcription factor that regulates anti-oxidative defenses, confers protection against CVD in animal models; however, when overactivated, it halts ROS physiological functions [57]. We evaluated the expression of protein KEAP-1 (Kelch-like ECH-associated protein 1), a sensor of oxidative stress that binds itself to Nrf2, promoting its degradation [58], and no differences in the cardiac tissue among experimental groups were found. Despite the absence of changes in the expression of KEAP-1, we cannot exclude oxidative imbalance. In order to elucidate whether endometriosis and metformin intervene in oxidative stress response in the heart, additional analysis, including evaluation of anti-oxidative enzymes expression and activity, will be necessary.

By aiming to elucidate the regulation of the expression of the studied molecules in the heart of mice and to identify putative markers of CVD risk in endometriosis, a selection of MIRs recognized to intervene in such processes was quantified.

MIR18a, MIR199 and MIR342 were appointed as serum markers of endometriosis [20,21,23]. In the heart of mice with endometriosis, however, the expression of MIR18a and of MIR199a was significantly lower, while MIR342 showed a decreasing trend, comparatively with groups S and M. These miRNAs intervene in cardiac cell survival mechanisms, and heart regenerative processes [24,25,27] and their reduced expression in mice with endometriosis indicates that the disease compromises the cardiac regenerative processes, which is supported by the increased cardiac fibrosis observed in such animals. In line with the fibrosis analysis data, metformin treatment had no effect on the expression of MIR18a, MIR199a and MIR342. The only MIR appointed as a serum marker of endometriosis [22] that showed a trend to increase in the heart with endometriosis was MIR320a. MIR320a contained in cardiomyocyte-released exosomes negatively affects cardiac repair and induces cardiomyocyte death [26].

Endometriosis downregulated the expression of MIR16-1, as evidenced in both groups E and ME relative to healthy animals. The MIR16 cluster family targets proangiogenic signaling pathways, including VEGF and VEGFR-2 gene expression, reducing local angiogenesis [27]. VEGF expression showed a trend to increase in animals with endometriosis, which aligns with the lessening of MIR16-1 expression.

Upregulation of MIR24-1, MIR200a and MIR20a is associated with cardioprotective mechanisms, mainly relating to apoptosis and fibrosis pathways regulation [25,28,29]. The reducing trend in the heart of mice with endometriosis may relate to poorer cardiac health. Overall, endometriosis seems to downregulate the expression of the cardioprotective MIRs, and metformin does not revert this condition.

On the contrary, expression of MIR155 that promotes cardiac hypertrophy by targeting calcium signaling [59] and inhibits angiogenesis [27] showed a slight trend to increase in metformin-treated relative to non-treated groups, which aligns with the higher cardiomyocyte areas found in group M.

The present study has some limitations regarding the translation of the results to the clinical setting. While the surgically induced mice model of endometriosis is a valuable tool to investigate pelvic and systemic features of endometriosis, rodents do not spontaneously develop the disease or menstruate.

## 5. Conclusions

To our understanding, this is the first study on a mice model that aims to elucidate cardiac morphologic alterations associated with endometriosis and the effect of metformin treatment. In summary, whilst treatment with metformin diminished endometriosis-associated endothelial dysfunction, it failed to mitigate NF-kB-associated inflammation and cardiac fibrosis. In addition, metformin-induced hypertrophy of the cardiomyocytes of mice. In order to fully understand whether the effect of metformin in the cardiac tissue of young mice with endometriosis is globally protected or otherwise deleterious, additional studies are necessary.

## Figures and Tables

**Figure 1 biomedicines-10-02782-f001:**
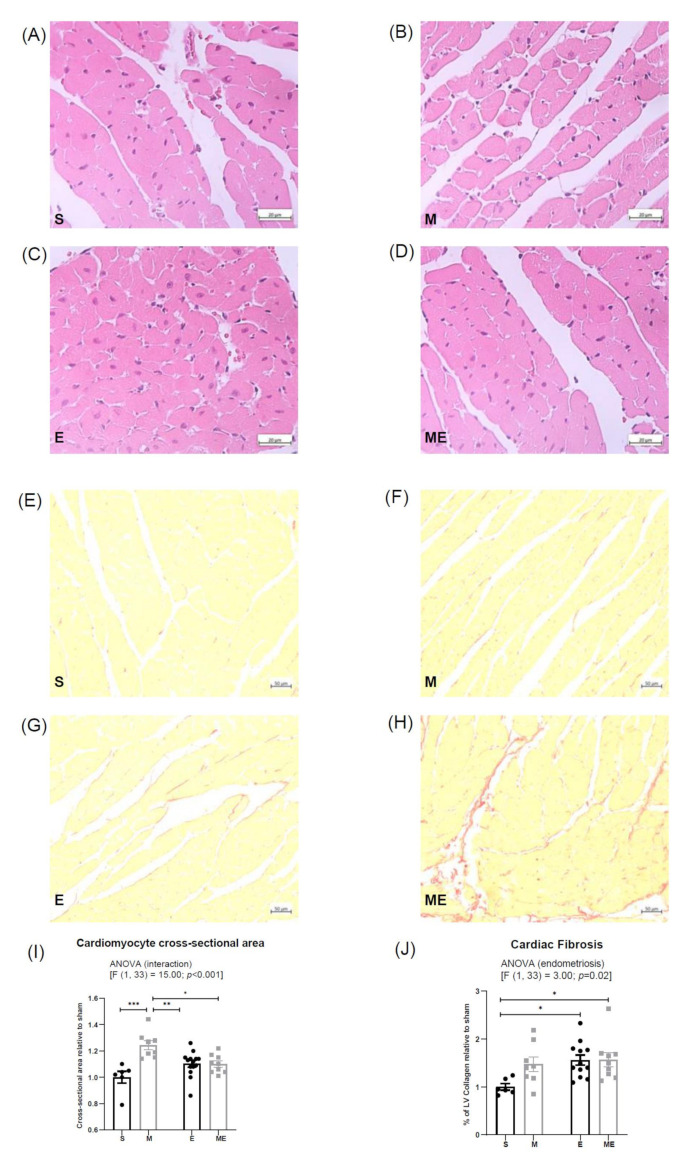
Representative images of heart sections stained with Hematoxylin–Eosin to assess cardiomyocyte area, groups S (**A**), M (**B**), E (**C**) and ME (**D**), and with Picrosirius Red to evaluate fibrosis, groups S (**E**), M (**F**), E (**G**) and ME (**H**). Graphical representation of cardiomyocytes cross-sectional surface area (**I**) and percentage of collagen on left ventricle (**J**) are shown. In each graphic, results in groups M, E and ME are displayed as fold increase relative to S group (represented by an arbitrary value of 1). n(S) = 6, n(M) = 8, n(E) = 12–14 and n(ME) = 9. Regular two-way ANOVA followed by *post-hoc* with Tukey’s test was employed. E—endometriosis, M—metformin, ME—metformin/endometriosis and S—Sham. Values displayed as mean ± SEM. * *p* value < 0.05, ** *p* value < 0.01, *** *p* value < 0.001.

**Figure 2 biomedicines-10-02782-f002:**
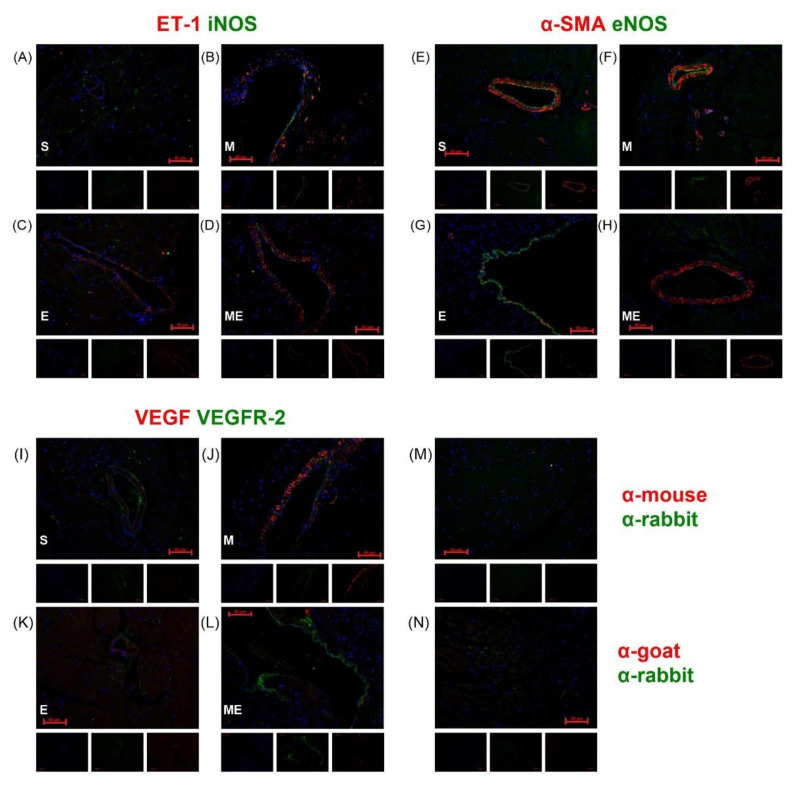
Representative images of dual-immunolabeling of ET-1(red) plus iNOS (green) in heart sections of groups S (**A**), M (**B**), E (**C**) and ME (**D**), of α-SMA (red) plus eNOS (green) in groups S (**E**), M (**F**), E (**G**), ME (**H**) and of VEGF (red) and VEGFR-2 (green) in groups S (**I**), M (**J**), E (**K**) and ME (**L**). Negative controls were prepared by omission of primary antibody (**M**,**N**). Nuclei were labeled blue (DAPI). The isolated blue, green and red channels are shown for all images. E—endometriosis, M—metformin, ME—metformin/endometriosis and S—Sham. DAPI—4′,6-diamidino-2-phenylindole, eNOS—endothelial nitric oxide synthase, ET-1—endothelin-1, iNOS—inducible nitric oxide synthase, α-SMA—alfa smooth muscle actin, VEGF—vascular endothelial growth factor and VEGFR-2—vascular endothelial growth factor receptor 2.

**Figure 3 biomedicines-10-02782-f003:**
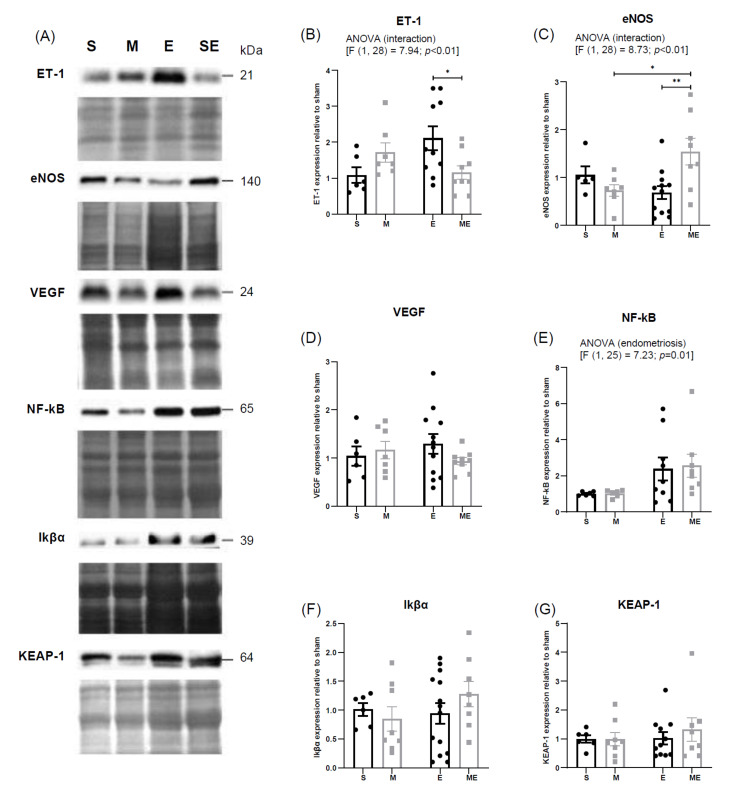
Representative bands of proteins obtained by Western blotting and respective Ponceau S staining (**A**). Graphical representation of expression levels of ET-1 (**B**), eNOS (**C**), VEGF (**D**), NF-κB (**E**), Ikβα (**F**) and KEAP-1 (**G**). Expression in groups M, E and ME is displayed as fold increase relative to S group (represented by an arbitrary value of 1). Values are displayed as mean ± SEM. * *p*-value below 0.05 and ** below 0.01. n(S) = 5–6, n(M) = 6–8, n(E)= 9–13 and n(ME) = 8–9. Regular two-way ANOVA followed by *post-hoc* with Tukey’s test and multiple *t*-tests per row with correction by the Holm–Sidak method was applied. E—endometriosis, M—metformin, ME—metformin/endometriosis and S—Sham. eNOS—endothelial nitric oxide synthase, ET-1—endothelin-1, Ikβα-kappa light polypeptide gene enhancer in B-cells inhibitor, alpha, KEAP-1—Kelch-like ECH associated protein 1, NF-kB—nuclear factor kappa-light-chain-enhancer of activated B cells, VEGF—vascular endothelial growth factor.

**Figure 4 biomedicines-10-02782-f004:**
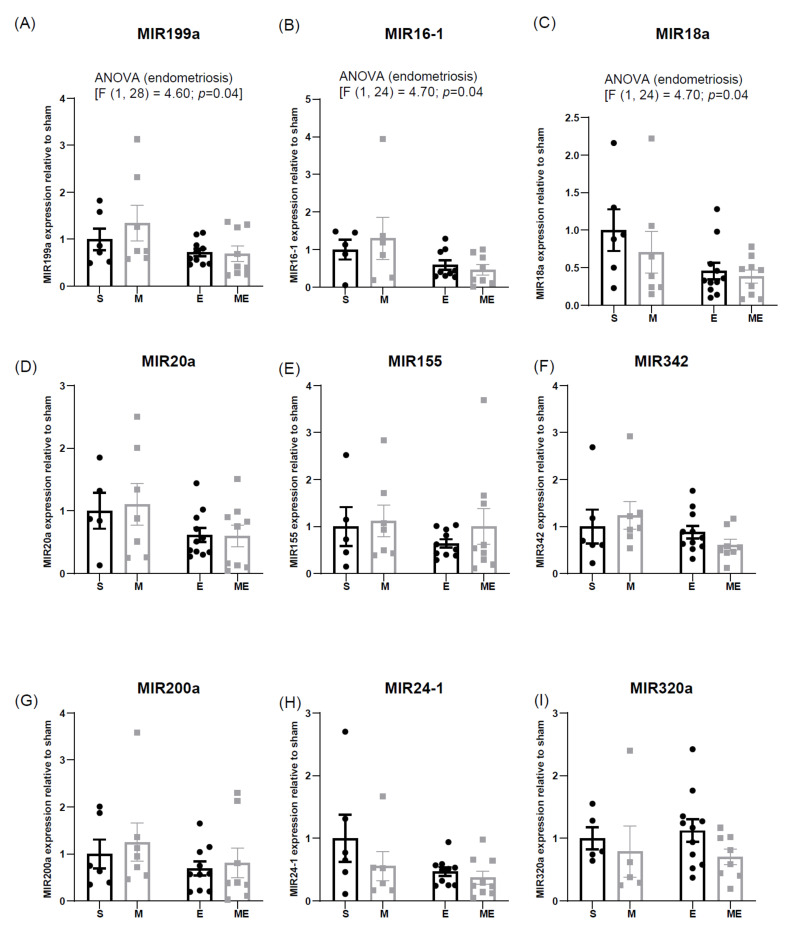
Graphical representation of expression levels of MIR199a (**A**), MIR16-1 (**B**), MIR18a (**C**), MIR20a (**D**), MIR155 (**E**), MIR-342 (**F**), MIR200a (**G**), MIR24-1 (**H**) and MIR320 (**I**) assessed by real-time qPCR. After employing the formula, 2^-(CT target miR – CT RNU)^, levels were normalized in groups M, E and ME relative to the expression of S group (represented by an expression value of 1). Values are displayed as mean ± SEM. n(S) = 5–6, n(M) = 5–7, n(E) = 9–11 and n(ME) = 8–9. Regular two-way ANOVA followed by *post-hoc* with Tukey’s test was employed. E—endometriosis, M—metformin, ME—metformin/endometriosis and S—Sham.

**Table 1 biomedicines-10-02782-t001:** List of antibodies used in immunofluorescence experiments.

Primary Antibodies	Concentration (v/v)
ET-1—Goat IgG Polyclonal-Santa Cruz Biotech, Dallas, Texas, USA. ref. sc-21625	1:100
VEGF—Goat IgG Polyclonal-R&D, Minneapolis, Minnesota, USA. ref. AF564	1:50
VEGFR-2—Rabbit IgG Polyclonal-Santa Cruz Biotech. ref. sc-504	1:100
iNOS—Rabbit IgG Polyclonal—Abcam Cambridge, UK. ref. ab178945	1:200
eNOS—Rabbit IgG Monoclonal-Santa Cruz Biotech. ref. sc-7271	1:200
α-SMA—Mouse IgG Monoclonal—Milllipore, Burlington, Massachusetts, USA. ref. CBL171	1:300
**Secondary Antibodies**	**Concentration (v/v)**
Anti-Goat Alexa Fluor 568 Molecular Probes Engene, Oregon, USA. ref. A11057	1:2000
Anti-Rabbit Alexa Fluor 488 Molecular Probes ref. A21206	1:2000
Anti-Mouse Alexa Fluor 568 Molecular Probes ref. A10037	1:2000

**Table 2 biomedicines-10-02782-t002:** List of antibodies used in Western Blotting.

Primary Antibodies	Concentration (v/v)	% Acrylamide Resolving Gel
ET-1—Rabbit IgG Polyclonal-R&D ref. sc-21625-R	1:1000	12
eNOS—Rabbit IgG Monoclonal-Cell Signaling Technology, Danvers, Massachusetts, USA. ref. D9A5L	1:1000	10
VEGF—Mouse IgG Monoclonal R&D ref. MAB564	1:100	14
NF-kB—Rabbit IgG Monoclonal-Cell Signaling Tech. ref. D14E12	1:1000	10
Ikβα—Mouse IgG Monoclonal-Cell Signaling Tech. ref. L35A5	1:1000	10
keap-1—Rabbit IgG Monoclonal-Cell Signaling Tech. ref. D6B12	1:1000	12
**Secondary Antibodies**		**Concentration (v/v)**
Goat Anti-Mouse IgG-HRP-Santa Cruz Biotech. ref. sc-2005		1:10,000
Donkey Anti-Rabbit IgG-HRP-Jackson ImmunoResearch, West Grove, Pennsylvania, USA. ref. 711-035-152		1:5000

**Table 3 biomedicines-10-02782-t003:** List of primers used in MIRs quantification by Real-Time qPCR.

MiRNA	Reference
RNU	Hs_RNU6-2_1 miScript Primer Assay (ref. MS00033740)
MIR199a	Hs_miR-199a_1 miScript Primer Assay (ref. MS00006741)
MIR16-1	has_miR-16-5p miRCURY LNA™ miRNA PCR Assay (ref. YP00205702)
MIR18a	has_miR-18a-5p miRCURY LNA™ miRNA PCR Assay (ref. YP00204207)
MIR20a	Hs_miR-20a_1 miScript Primer Assay (ref. MS00003199)
MIR155	Mm_miR-155_1 miScript Primer Assay (ref. MS00001701)
MIR342	has_miR-342-3p miRCURY LNA™ miRNA PCR Assay (ref. YP00205625)
MIR200a	Hs_miR-200a_1 miScript Primer Assay (ref. MS00003738)
MIR24-1	has_miR-24-3p miRCURY LNA™ miRNA PCR Assay (ref. YP00204260)
MIR320a	Hs_miR-320a_1 miScript Primer Assay (ref. MS00014707)

## Data Availability

The datasets generated during and/or analyzed during the current study are available from the corresponding author upon reasonable request.

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
