# Peer review of "Metformin Prevents Endothelial Dysfunction in Endometriosis through Downregulation of ET-1 and Upregulation of eNOS"

_biomedicines, 2022, doi:10.3390/biomedicines10112782_

Round 1

Reviewer 1 Report

The present study determined the efficacy of metformin in reducing the CVD risk associated with endometriosis. Pathways associated with endothelial function, angiogenesis, inflammation, and oxidative stress were studied in the mouse model of endometriosis and changes in morphology and the expression of the most important microRNAs implicated in endometriosis and heart injury were also examined. The authors draw a conclusion that this is the first study using a mouse model to investigate endometriosis-related cardiac changes and the effect of metformin treatment. This reviewer has the following concerns and suggestions:

Major comments

1.       There are lots of results in the study, but the conceptual novelty appears to be limited. The authors emphasized that this is the first study to clarify cardiac changes related to endometriosis and the impact of metformin therapy. However, other previous studies already indicated an association between endometriosis and the risk of coronary heart diseases (PMID: 27025928; PMID: 33683770; PMID: 25935777) while a recent study has shown that endometriosis promotes atherosclerosis in ApoE-/- mice (PMID: 35351413).

2.       It is insufficient to detect the expression of Keap1 to illustrate the oxidation-responsive pathways. The authors shall examine the expression of Nrf2 and its target genes (HMOX1, NQO1), superoxide dismutase (SOD) and catalase using western blotting or RT-PCR.

3.       In Figure 2, the authors just displayed the cell localization of ET-1, VEGF, VEGFR-2, eNOS, iNOS in the endothelium or vascular smooth muscle cells in the heart. However, their distributions are well-known. This reviewer suggests the authors providing statistical analysis results for Figure 2 to further prove that metformin indeed decreases the expression of ET-1 and increases eNOS expression as the mechanism for improved endothelial function.

4.       In line 393-395, “metformin failed to demonstrate an anti-inflammatory effect …”. This conclusion is overstated because the anti-inflammatory effect is not only regulated by NF-κB signalling. For example, the anti-inflammatory effect of metformin can be also achieved by activating AMPK and Nrf2 (PMID: 25772174; PMID: 25794703; PMID: 33584319). The discussion needs update.

5.       In lines 401-405, the authors stated that metformin administration to healthy animals resulted in a significant increase in the size of cardiomyocytes and induced mild fibrosis. Please cite the literature and discuss more about the mechanisms by which metformin is reported to induce myocardial hypertrophy.

Minor points

1.     In the Method section 2.6, the authors stated that “Results are presented as mean±SEM”, but in the figure legend, the results are presented as “mean ± standard error”, the authors should make sure the information is correct about data presentation.

2.     The amplification of immunofluorescence images in Figure 2 is inconsistent, especially in Figure 2a, Figure 2g and 2i. Please provide statistical analysis results for Figure 2.  

3.     In Figure 3, the quality of Ik-βα WB data is low. The obvious change in the width of the band implies that the protein content loaded may not be the same. I recommend that the authors provide all the original WB analysis data to ensure that the data is properly analyzed and presented.

Reviewer 2 Report

The purpose of the paper was to evaluate the role of metformin in the mitigation of CVD 

risk associated with endometriosis. 

That is very interesting paper, however I have a few questions: 

  1. In the introduction, could you also explain the coagulation problems in women with endometriosis. It also may increase the risk of cardiovascular disease. 

  1. How hormone therapy may influence the development of these diseases? 

  1. Why did researchers use this type of treatment?  

  1. Can researchers provide studies to support attempts to use metformin to treat endometriosis? 

  1. Can the type of endometriosis, endometrioma, DIE or peritoneal, affect the test results? 

  1. What practical application could the results of these studies have?
